environmental science, ecology

agroecosystem, *Bombus impatiens*, crop pollination, herbicide, insecticide, pollinator health

**Author for correspondence:**
Ian Kaplan
e-mail: ikaplan@purdue.edu

†These authors contributed equally to this study.

Electronic supplementary material is available online https://doi.org/10.6084/m9.figshare.c.5469506.

# Supplemental forage ameliorates the negative impact of insecticides on bumblebees in a pollinator-dependent crop

Laura L. Ingwell[1,†], John J. Ternest[1,2,†], Jacob R. Pecenka[1] and Ian Kaplan[1]

[1]Department of Entomology, Purdue University, 901 West State Street, West Lafayette, IN, USA
[2]Department of Entomology and Nematology, University of Florida, 1881 Natural Area Drive, Gainesville, FL, USA

IK, 0000-0003-4469-2750

Insecticide use and insufficient forage are two of the leading stressors to pollinators in agroecosystems. While these factors have been well studied individually, the experimental designs do not reflect real-world conditions where insecticide exposure and lack of forage occur simultaneously and could interactively suppress pollinator health. Using outdoor enclosures, we tested the effects of insecticides (imidacloprid + lambda-cyhalothrin) and non-crop forage (clover) in a factorial design, measuring the survival, behaviour and performance of bumblebees (*Bombus impatiens*), as well as pollination of the focal crop, watermelon. Colony survival was synergistically reduced to 17% in watermelon alone + insecticides (survival was 100% in all other treatments). However, behavioural shifts in foraging were mainly owing to insecticides (e.g. 95% reduced visitation rate to watermelon flowers), while impacts on hive performance were primarily driven by clover presence (e.g. 374% increase in the number of live eggs). Insecticide-mediated reductions in foraging decreased crop pollination (fruit set) by 32%. Altogether, these data indicate that both insecticides and non-crop forage play integral roles in shaping pollinator health in agricultural landscapes, but the relative importance and interaction of these two factors depend on which aspect of 'health' is being considered.

## 1. Introduction

Uncovering the factors causing declines in pollinator health is complicated by the fact that environmental stressors probably act in concert rather than individually. Thus, efforts should be made to experimentally test the leading stressors both alone and in combination to tease apart the main and interactive effects. For bees, most experts agree that two of the primary causes underlying population declines or reductions in performance are insecticide exposure and lack of forage [1]. Insecticidal toxicity is largely driven by the widespread integration of neonicotinoid seed treatments, which has dramatically elevated the toxic load for bees across agricultural landscapes, despite lower overall amounts of product applied [2,3]. The lack of forage is a result of multiple factors surrounding the general phenomenon of agricultural intensification, e.g. conversion of once-diverse prairies into monocultures of one or a few flowering crops; increased herbicide inputs, such as glyphosate and dicamba, owing to the adoption of herbicide-tolerant transgenic crops. Collectively, this process has sterilized landscapes by removing non-crop flowering plants, often with negative outcomes for bees inhabiting these regions [4–7].

The two mechanisms, however, are not independent. Insecticide-mediated toxicity is likely to interact with changes to the flowering plant community in agroecosystems for several reasons. First, non-crop pollen is expected to

dilute the higher insecticide load derived from a monofloral crop pollen diet, thereby acting as a toxicity buffer. This assumes that crop pollen contains more toxic insecticides than non-crop pollen, which is not always the case [8,9]. Second, a lack of non-crop flowers can funnel pollinators into crop fields in search of food, increasing spatial overlap with areas targeted for insecticide application. Last, a lack of non-crop flowers reduces pollen quantity and diversity, both of which can make bees more stressed and physiologically vulnerable to insecticide toxicity [10–12].

Regardless of mechanism, the general prediction is that reduced access to non-crop flowers will exacerbate the detrimental effect of insecticides on pollinators. A corollary of this prediction is that managing herbicide regimes, land use and/or wildflower habitat can attenuate the negative impact of insecticides. The presence of natural habitat in the landscape is known to buffer the negative effects of insecticide use [13,14]. However, few studies have experimentally manipulated supplemental forage and insecticide use in tandem (but see [15,16]), and these mostly employ controlled laboratory experiments that do not resemble a natural foraging arena [17,18]. Virtually, all of the existing work in this area also employs artificial feeders that titrate insecticides or nutritional resources via sugar water rather than simulating actual exposure routes and concentrations that a bee encounters while foraging in the field [19].

Furthermore, the interactive effects of non-crop flowers and insecticides, at present, are only studied from the perspective of pollinator health; we know far less about consequences for crop pollination, even though the two, in theory, should go hand in hand (i.e. healthier pollinators are expected to be more effective pollinators). Insecticide studies in general tend to only measure pollinator performance, with few simultaneously quantifying crop pollination in realistic, outdoor environments [20]. Similarly, the impacts of supplemental forage for crop pollination are complicated. Increased availability of non-crop flowers is assumed to benefit pollinator nutrition and thus performance, but this benefit may not extend to crop pollination [21,22]. A widespread concern among growers is that heterospecific pollen will result in stigma clogging, whereby bees deposit wild plant pollen onto crop flowers, reducing yield. Another worry is that supplemental forage is a distraction that will 'pull' bees away from the focal crop, reducing visitation rates during periods of peak bloom when they are most needed. The likelihood for these outcomes depends on factors such as phenological synchrony between non-crop and crop plants [23], and the degree to which pollinators overlap in foraging across adjacent habitats, which is high for polylectic managed bees but could be low for oligolectic wild taxa [24].

Here, we use large, semi-field enclosures to evaluate the singular and combined impacts of insecticides and clover presence as a supplemental forage on: (i) the survival, performance and foraging behaviours of managed bumblebees (Bombus impatiens); and (ii) pollination of the focal crop, watermelon (Citrullus lanatus) (electronic supplementary material, figure S1). Bumblebees, while less popular than honeybees, are commonly used as a secondary pollinator on commercial watermelon farms [25]. This species is also highly sensitive to insecticides applied to crops with much of the recent emphasis on systemic neonicotinoids [26–30] and responds favourably to flowering resource availability [31]. We tested the hypothesis that insecticide use has a comparatively stronger detrimental effect on bees and, correspondingly, crop pollination in the absence of non-crop forage.

## 2. Methods

### (a) Site description and plant propagation

Experiments were conducted at the Meigs Horticultural Farm, part of the Throckmorton Purdue Agricultural Center (TPAC), located in Lafayette, IN, USA. Six high tunnels were used as field cages to control the foraging radius of experimental bumblebee hives. Each tunnel measured 14.6 × 7.9 × 3.7 m; length, width, height (LWH) and was covered in a single layer of plastic allowing 80% light transmission (12-mil ClearSpan™ PolyMax®). Openings were covered with insect exclusion screening (Anti-Insect Netting, 25 Mesh, Greenhouse Megastore, Danville, IL, USA) to keep bumblebees inside the tunnel. In addition, each tunnel was divided in half, lengthwise, using exclusion screens to separate the tunnel into two arenas. There were a total of 12 arenas used in this experiment, each measuring 7.3 × 7.9 × 3.7 m LWH.

The focal crop used in this system was watermelon, which is highly reliant on managed bees for pollination, requiring at least 12 bumblebee visits per flower for optimal yield [25,32,33]. We used a diploid pollenizer variety, AcePlus, to optimize flower availability within the limited foraging space. See the electronic supplementary material, Methods for additional detail on crop and experiment management.

### (b) Experimental design

The experiment employed a 2 × 2 factorial design with two levels of insecticide use (±) and two levels of non-crop forage (±), resulting in the following four treatments: (i) untreated watermelon [insecticide (−)/non-crop forage (−)]; (ii) insecticide-treated watermelon [insecticide (+)/non-crop forage (−)]; (iii) untreated watermelon with clover [insecticide (−)/non-crop forage (+)]; and (iv) insecticide-treated watermelon with clover [insecticide (+)/non-crop forage (+)]. Each of these treatments was assigned to one of the 12 caged arenas. The experiment was organized in a split-plot design where insecticide was the main plot factor applied to the whole tunnel and forage was the subplot factor nested within the main plot using bisected tunnels on either side of the exclusion netting (i.e. one half was seeded with forage; the other was not). For a visual depiction, see the electronic supplementary material, figure S2. A pair of adjacent tunnels (east–west orientation) consisted of a single block, replicated three times along a north–south gradient. The experiment was conducted twice (once in May–July 2018 and again in July–September 2018), resulting in a total of six replicates of each treatment combination.

In arenas assigned to non-crop forage, we directly seeded (15 kg ha$^{-1}$; 1–2 cm depth) in October 2017 and April 2018 with a clover mix consisting of 44% yellow blossom sweet clover (Melilotus officinalis), 33% red clover (Trifolium pratense) and 23% Ladino white clover (Trifolium repens) (3-Way Clover Mix; King's AgriSeeds, Ronks, PA, USA). This seed mix is marketed as a pollinator-friendly blend and clovers, in general, are among the most nutritionally important forages for bumblebees [34,35].

In the insecticide (+) treatment, a systemic neonicotinoid was applied as a root drench to the soil at transplant. To do so, watermelon seedlings received imidacloprid (Admire Pro®) at a rate of 730.5 ml ha$^{-1}$ by watering each seedling with 500 ml of the insecticide solution. This concentration was based on the recommended label rate for Admire Pro used by growers of 511.6–768.9 ml ha$^{-1}$. In addition, the pyrethroid lambda-cyhalothrin (Warrior II with Zeon Technology® at 140.2 ml ha$^{-1}$) was applied to the watermelon plants using an electrostatic sprayer four weeks post-transplant.

This combination of neonicotinoid at planting, followed by a subsequent foliar pyrethroid spray, is a commonly used insecticide regime targeting striped cucumber beetles (*Acalymma vittatum*) and other insect pests by watermelon growers in the midwestern USA [36].

## (c) Bumblebee performance and crop pollination

When watermelon vines began flowering (*ca* two weeks after transplant), each arena received one bumblebee hive (*B. impatiens*; Koppert Biological Systems, Howell, MI, USA). Excel research hives were used, which contain a minimum of 70 workers, one queen and an observation lid. Although nectar sacks were installed in the hives, we did not remove the lid and thus bees were unable to access this resource. Similarly, pollen patties were not added to hives. As a result, observed changes in hive survival and performance are a direct reflection of the experimentally imposed treatments and not buffered by supplemental nutrients. We also assumed this would encourage bees to forage on flowers provided in the arenas. Nest covering material was excluded so we could observe hives throughout the experiment.

Hives were placed on 25 May and 15 August where they remained for six weeks. Within arenas, hives were positioned in a corner, elevated from the soil on a plastic crate and shaded beneath an umbrella. The entire hive box (nest-box + nectar + outer cardboard box) was weighed immediately prior to placement in arenas and again at the end of the six-week foraging period to estimate weight gain (final − initial weight) as a measure of performance. Hives were also checked daily to track survival over time. They were considered dead when no active bees were observed tending the colony or out foraging on flowers for two consecutive surveys. In the first trial, one hive in the insecticide (−)/non-crop forage (−) treatment was knocked over owing to high wind and died soon after placement (29 May). This replicate was excluded from statistical analyses of all bumblebee and crop variables, resulting in five replicates rather than six.

Forager surveys were conducted twice per week for the six-week duration of bumblebee placement. During surveys, observers walked each row in a transect fashion, recording the number of total foragers active and identity of the flower that each was visiting (i.e. watermelon versus clover). At the end of observations, the hive lid was opened and the number of bees working inside the hive was counted. The queen, if visible, was recorded as dead or alive and the presence of honeypots or pollen stores was also noted.

Pollination efficiency was evaluated by flagging 15 female watermelon flowers in each arena (11 June in trial 1; 27 August in trial 2) and following those flowers for one week to verify if pollination had occurred. This was determined by measuring the presence and size of the developing fruit (electronic supplementary material, figure S3). Flagged fruits were then tracked through maturity and eventually weighed to calculate individual melon weights; however, fruit yield data were only measured in trial 2.

We monitored within-hive behaviour during each trial on 5 July and 12 September. GoPro cameras (Hero 5) were placed in each arena on a tripod focused directly over the observation lid, draped with a white cloth for shade. Videos were analysed for hive activity using EthoVision software. See the electronic supplementary material, Methods for protocol details.

At the conclusion of each experiment or upon colony death, nest-boxes were placed in a freezer. In the laboratory, we dissected each hive, recording the number and weight of workers and queens. Weights were calculated by combining all bees from each group (worker or queen), measuring total weight and dividing by the number of individuals, resulting in a single average value per hive. We also recorded: the number of open worker cells and open queen cells (distinguished by size;

see the electronic supplementary material, figure S4); number of worker cells filled with nectar; number of live egg cells and number of dead larval cells. These variables are commonly used to estimate bumblebee hive health in response to insecticides and other stressors [30,37]. Eggs were considered dead when they were desiccated and/or black upon visual inspection. Viable eggs are milky in colour, oval-shaped and free of secondary pathogens.

## (d) Pesticide residue sampling and quantification

Soil, clover and watermelon flowers, and bee nest material were collected to measure the residual amount of insecticides present in each arena. Sampling and analytical details are provided in the electronic supplementary material, Methods.

## (e) Data analysis

Survival analysis was performed using the duration of colony survival in days since placement in the foraging arenas. All hives that survived the entire six-week trial period were right-censored. The Kaplan–Meier estimates were used to create survival curves and a log-rank test was performed to compare among the four treatments.

For bee foraging data, we only included observations when hives survived across all experimental arenas (i.e. we censored data from dates after hive death began). To avoid pseudoreplication and provide a single value per arena, we summed foraging observations collected over multiple dates. The effects of the two focal treatments on bee foraging behaviour were analysed using negative binomial regression because count data were overdispersed. For watermelon observations, we tested the main and interactive effects of insecticide and non-crop forage; however, for clover, we only included the effect of insecticide in the non-crop forage (+) treatment because there was no clover to observe in the (−) treatment. For both flower types, we also included trial (1 versus 2) and spatial block (high tunnels 1–2, 3–4, 5–6) as predictor variables.

For hive activity and colony performance, main and interactive effects of insecticide and non-crop forage were tested for each response. Continuous data were analysed using a normal distribution for final hive weight, while a $\log(x + 1)$ transformation was used for queen and worker weights to improve normality. Discrete count data (worker no., worker larvae, worker pupae, worker honeypots, dead larval cells, live eggs) were analysed with a negative binomial or zero-inflated negative binomial model. As with foraging data, we included trial (1 versus 2) and spatial block (high tunnels 1–2, 3–4, 5–6) as predictor variables. We also included initial colony weight as a covariate.

To assess fruit set, we performed an arcsine square root transformation on the proportion of flagged flowers that developed into a fruit per experimental arena. Insecticide, non-crop forage, trial and block were used as predictor variables using a general linear model. The mean fruit weight (kg) per melon per tunnel was analysed with a regression model using a lognormal distribution; we did not include trial as a factor as fruit weights were only measured in trial 2.

All statistical analyses were conducted using JMP Pro 15 (SAS Institute Inc., Cary, NC, USA).

## 3. Results

### (a) Bumblebee survival

In both treatments containing clover (insecticide ±), bumblebee hive survival was 100% over the full six-week experiment period, regardless of pesticide application. In the absence of clover, however, survival depended on whether the crop was treated with insecticides; namely, in the insecticide (−)

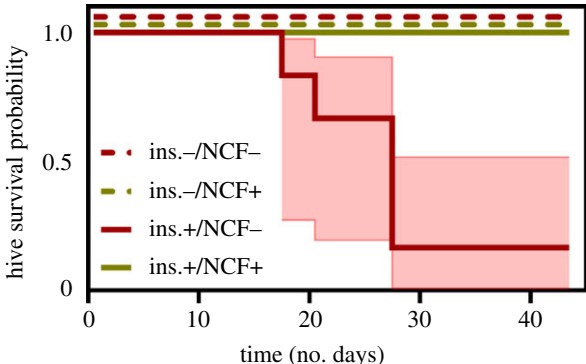

**Figure 1.** Survival curves with 95% confidence interval showing the persistence of bumblebee hives in the presence/absence of insecticides and alternative non-crop forage in the four treatment combinations. Hives remained in arenas for six weeks and colony survival was assessed daily. Lines at 1.0 were jittered for visibility. Ins., insecticide; NCF, non-crop forage (clover). (Online version in colour.)

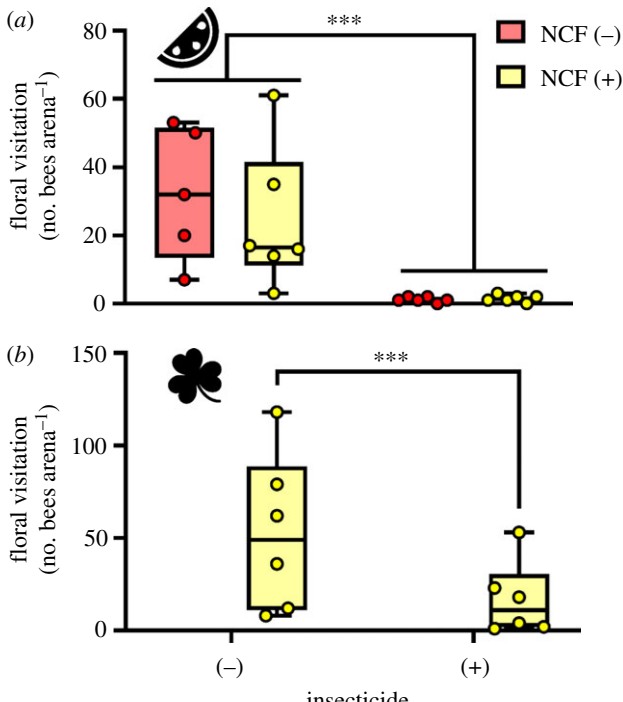

**Figure 2.** Visitation rate of bumblebees to flowers from observational transect walks, including watermelon (*a*) and clover (*b*) flowers. Box plots show the median, quartiles, range and individual data points across the four treatments. NCF, non-crop forage (clover). \*\*\*$p < 0.001$. (Online version in colour.)

treatment all hives survived, while in the insecticide (+) treatment, survival was reduced to 17% (figure 1; log-rank test, $\chi_3^2 = 19.8$, $p = 0.0002$).

## (b) Bumblebee foraging and in-hive behaviours

Insecticides reduced bee visitation to watermelon flowers by 95% compared with untreated control arenas (insecticide (−) versus (+), mean = 28.0 and 1.3 observations arena$^{-1}$, respectively), but clover presence did not affect watermelon visitation rate (figure 2*a* and table 1*a*). Similarly, insecticides caused a 68% reduction in foraging on clover (insecticide (−) versus (+), mean = 52.5 and 16.8 observations arena$^{-1}$, respectively), despite only being applied to the crop

(figure 2*b*). Importantly, we found no difference in bee foraging rates ($p = 0.9040$) comparing early observation periods before pyrethroids were applied as a foliar spray in the insecticide (+) treatments (i.e. isolating only the effect of the systemic neonicotinoid at-planting) with later dates after the watermelon was sprayed.

EthoVision analysis of video recordings from within-hive activity also revealed a strong negative influence of insecticides with no corresponding impact of clover (figure 3 and table 1*b*). Bees from insecticide-exposed hives exhibited a 57% reduction in activity (insecticide (−) versus (+), mean = 1.9 and 0.8% activity within observation area, respectively).

## (c) Bumblebee colony performance

Unlike foraging behaviour, colony performance showed the opposite pattern with stronger overall effects of clover than insecticides (table 1*c*). Insecticide use increased the number of worker honeypots (insecticide (−) versus (+), mean = 0.09 and 19.3 hive$^{-1}$, respectively), while decreasing the number of live eggs (insecticide (−) versus (+), mean = 22.18 and 10.5, respectively). Clover presence increased worker weight (clover (−) versus (+), mean = 0.92 and 3.23 g, respectively) and number of live eggs (clover (−) versus (+), mean = 5.45 and 25.83, respectively), but decreased the number of worker pupae (clover (−) versus (+), mean = 55.91 and 20.17, respectively) and dead larval cells (clover (−) versus (+), mean = 47.64 and 22.25, respectively) (means for all treatments reported in the electronic supplementary material, table S2). Two variables—queen weight and worker larvae—were affected by the interaction between insecticide use and clover presence. Hive weight and worker count were the only variables unaffected by either factor. Data on number of reproductives (queens and males) were excluded as they were always either zero or one per hive. No new queens were produced by the colonies in this experiment (i.e. in cases where one was found, it was probably the old mother queen originating with the hive).

## (d) Crop pollination

The mean fruit set (proportion of female flowers producing fruit) was 0.31 across all treatments combined ($n = 360$ flowers); this rate is normal for watermelon, which ranges from 0.2 to 0.4, even under optimal conditions [38]. Insecticides reduced fruit set by 32% (insecticide (−) versus (+), mean = 0.37 and 0.25, respectively) with no corresponding effect owing to clover presence (figure 4 and table 1*d*). By contrast, individual fruit weights (kg) were unaffected by insecticide use but were reduced by 21% owing to clover presence (clover (−) versus (+), mean ± s.e. = 1.51 ± 0.09 and 1.20 ± 0.04, respectively). Because fruit set was not significantly different, and trended towards being slightly lower, in the presence of clover, changes in this variable could not compensate for the decline in individual fruit weight to affect total fruit weight per arena.

## (e) Pesticide residues

The neonicotinoid imidacloprid was detected in all matrices in which it was tested, including soil, watermelon pollen, clover pollen and bumblebee nest material (summary data reported in the electronic supplementary material, table S3). In virtually all cases, detection rates increased dramatically

**Table 1.** Statistical table describing the main effects of insecticide and non-crop forage (clover) on all bumblebee and crop response variables (d.f. = 1 throughout). (Insecticide × NCF denotes statistical interaction between the two main effects. Significant ($p < 0.05$) and marginally significant ($p < 0.07$) insecticide and non-crop forage effects are italicized for emphasis. Yellow and red shading denote increases and decreases, respectively, of the response variable to treatments that were significant for main effects (note a few interactions were also significant; left unshaded). Effects of non-treatment-related variables (e.g. block, trial, initial hive weight) are reported in the electronic supplementary material, table S1.

| | insecticide | | non-crop forage (NCF) | | insecticide × NCF | |
|---|---|---|---|---|---|---|
| | $\chi^2$ | *p*-value | $\chi^2$ | *p*-value | $\chi^2$ | *p*-value |
| (*a*) floral visitation rate | | | | | | |
| watermelon | *54.70* | *<0.0001* | 0.28 | 0.5994 | 0.67 | 0.4122 |
| clover | *17.64* | *<0.0001* | n.a. | n.a. | n.a. | n.a. |
| (*b*) in-hive activity | *5.68* | *0.0172* | 1.91 | 0.1673 | 0.34 | 0.5565 |
| (*c*) colony performance | | | | | | |
| hive weight | 2.80 | 0.0941 | 0.12 | 0.7321 | 1.81 | 0.1779 |
| queen weight | 1.32 | 0.2502 | *19.98* | *<0.0001* | 5.52 | 0.0188 |
| worker weight | *3.46* | *0.0628* | *9.16* | *0.0025* | 1.49 | 0.2213 |
| worker count | 0.03 | 0.8608 | *3.64* | *0.0565* | 0.66 | 0.4160 |
| worker larvae | *3.65* | *0.0561* | *16.56* | *<0.0001* | 4.91 | 0.0268 |
| worker pupae | 0.37 | 0.5412 | *5.85* | *0.0155* | 0.08 | 0.7801 |
| worker honeypots | *18.30* | *<0.0001* | *6.14* | *0.0132* | 0.02 | 0.8861 |
| dead larval cells | 0.75 | 0.3848 | *6.39* | *0.0115* | 0.10 | 0.7547 |
| live eggs | *5.01* | *0.0253* | *4.38* | *0.0364* | 0.00 | 0.9601 |
| (*d*) pollination | | | | | | |
| fruit set | *6.32* | *0.0119* | 2.02 | 0.1550 | 1.08 | 0.2993 |
| fruit weight | 2.18 | 0.1395 | *6.95* | *0.0084* | 2.98 | 0.0844 |

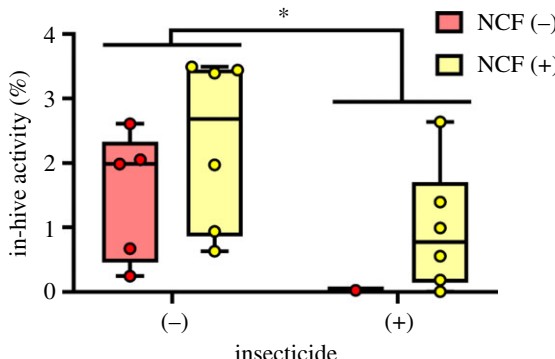

**Figure 3.** Within-hive bee activity from ETHOVISION analysis of video recordings. Insecticide use significantly reduced activity, whereas non-crop forage had no significant effect. GoPro cameras focused directly over the observation lid recorded for at least 20 min with activity (%) quantified by measuring the amount of pixel change in the videos over time. The arena size for each analysis was standardized to a 9.5 × 7 cm area focused over the largest portion of brood in the hive. Box plots show the median, quartiles, range and individual data points across the four treatments. NCF, non-crop forage (clover). *$p < 0.05$. (Online version in colour.)

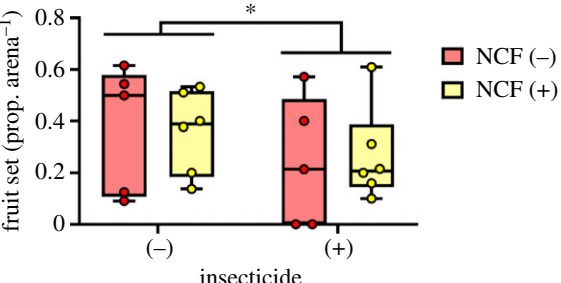

**Figure 4.** Fruit set of tagged watermelon flowers in the four treatments. Insecticide use significantly reduced fruit set, whereas non-crop forage had no significant effect. Fruit set was calculated as the proportion of flowers developing into a fruit per experimental arena ($n = 15$ flowers arena$^{-1}$) and used as a measure of pollination. Box plots show the median, quartiles, range and individual data points across the four treatments. NCF, non-crop forage (clover). *$p < 0.05$. (Online version in colour.)

with insecticide use; for example, detection in watermelon pollen ranged from 0 to 20% in the insecticide (−) treatment and increased to 92–100% in the insecticide (+) treatment. It is notable that imidacloprid was detected at high rates in clover pollen (100%) and nest material (80–83%) since it was not applied to these areas. However, the mean concentrations were two to three times higher in crop than non-crop pollen. Overall, the highest concentrations were found in the soil (max. 5574 ng g$^{-1}$), followed by pollen (max. 377 ng g$^{-1}$), and lowest levels were in nest material (max. 8 ng g$^{-1}$). The related neonicotinoids, thiamethoxam and clothianidin were also measured in samples, but were mostly below the limit of detection (thiamethoxam: soil—0% samples detected, pollen—3% samples detected (median = 0.99 ng g$^{-1}$, max. = 1.36 ng g$^{-1}$); clothianidin: soil—50% samples detected (median = 0.77 ng g$^{-1}$, max. = 3.59 ng g$^{-1}$), pollen—1% samples detected (1.46 ng g$^{-1}$ in a single sample)).

# 4. Discussion

Overall, we found partial support for our hypothesis that insecticides have stronger negative effects on bees and crop pollination in the absence of non-crop forage. The strongest evidence came from hive survival, where insecticide use and lack of forage synergistically increased mortality. It was particularly striking that survival was 100% in all three treatments with zero or one stressor alone, but the combination of two stressors caused survival to plummet to 17%. These data suggest that cultivating or encouraging supplemental flowers in and around crop fields can buffer the negative non-target effects from insecticides [14,16]. In the field, growers could accomplish this by implementing wildflower strips along field borders or adopting a less stringent herbicide regime to encourage flowering weeds. Alternatively, investing in integrated pest management (IPM) programmes or technologies that reduce insecticide use (e.g. pest scouting and action thresholds; host plant resistance) could generate the same outcome.

We were unable to pinpoint the mechanism underlying this synergy. However, it is unlikely a behavioural result of 'forcing' bees to forage on the insecticide-treated crop owing to a lack of alternatives, because clover presence did not affect the visitation rate on watermelon flowers. Watermelon is not a preferred resource for bumblebees in the field [32,33,39], but in our experiment, they readily visited these flowers even when surrounded by clover, which is considered a high-quality forage whose flowers are open and accessible during the same time of day as watermelon (i.e. temporal coincidence in bloom times). Clover flowers also contained non-trivial amounts of imidacloprid, albeit in lower concentrations than watermelon. This means that dietary insecticide exposure would have occurred, even if bees exclusively foraged from clover. The detection of imidacloprid in neighbouring clover is not surprising, given their close proximity (1–2 m) to treated watermelon in arenas and the propensity for water-soluble neonicotinoids to move laterally in the soil profile. A more likely explanation for the synergistic reduction in bumblebee survival is that co-exposure to both stressors acted at a physiological level to increase vulnerability. This stressor combination may have been exacerbated by the fact that screened tunnels are considerably warmer than ambient conditions owing to reduced ventilation, resulting in heat stress as a likely contributing factor. Moreover, we removed nest covering material to observe hives, which could have further impeded thermoregulation. The temporal pattern of hive death also supports this explanation. Several hives died near-simultaneously within a few days in early September. The 5 days immediately preceding their death experienced the highest daily maximum temperatures of the entire experiment (avg. 41.2°C; for comparison, the two weeks prior to this (16–31 August) had maximum temperatures that were greater than 5° cooler (avg. 35.1°C)). Interestingly, a recent study demonstrated that poor diet quality increases susceptibility of the bumblebee *Bombus terrestris* to heat waves [40], and bumblebees in general seem to respond poorly to warm environments [41]. Experimentally testing forage/insecticide interactions under a range of temperatures is necessary to tease apart these relationships.

In our study, the specific route of insecticide exposure for bees is unknown. Imidacloprid was detected in all matrices tested (soil, pollen, nest material) with highest levels in soil.

This is not particularly surprising since the insecticide was applied as a soil drench, rather than a foliar spray, and the product moved systemically through the plant to reach flowers. As a result, soil exposure is likely to pose a threat to ground-nesting wild bees that come in close contact with these residues [42–44]; however, given that bumblebees were housed in aboveground structures in the experiment, we assume that insecticide detected in nest materials derived from oral exposure via collection of contaminated floral resources. The specific concentrations reported for imidacloprid in crop-treated pollen were high (median in watermelon: 75.4 ng g$^{-1}$) but within the range detected in other studies of neonicotinoids in cucurbits [45–47]. For example, Bloom *et al.* [45] reported the median thiamethoxam concentration in seed-treated cucumber pollen from commercial fields at 73.7 ng g$^{-1}$. Similarly, Dively & Kamel [46] found the mean imidacloprid residues in pumpkin pollen at 60.9 ng g$^{-1}$, using the same soil drench technique and product used here. Because insecticides were applied in covered high tunnels in our study, the residues measured could differ from open-field values. Protected structures prevent rain downpour from leaching pesticides from the soil and the plastic covering filters sunlight, potentially leading to altered pesticide degradation rates. We also detected imidacloprid in untreated watermelon and clover flowers, which could underestimate the effect of insecticides if the control is contaminated. This off-site detection is not surprising because: (i) imidacloprid is among the commonly used insecticides in agricultural areas where the study was conducted; and (ii) our analytical approach using liquid chromatography-mass spectrometry (LC-MS) is highly sensitive to trace amounts that, while detectable, may not be biologically relevant. For context, the median value of imidacloprid in treated flowers was *ca* 250 and 50 times higher than untreated flowers for watermelon and clover, respectively. Thus, we are confident that bees in the untreated arenas were not inadvertently exposed to high levels of imidacloprid.

An additional factor strongly affecting our interpretation of forage/insecticide relationships is the response variable measured. While survival clearly showed a synergistic reduction, the other variables were dominated by main effects of either forage availability or insecticides. These main effects were remarkably consistent with short-term behaviours, such as in-hive activity and flower visitation, negatively affected by insecticides. Reductions to in-hive activity were expected based on the fact that imidacloprid and other neonicotinoids are neurotoxins that impede motor function in bees [48]. Despite the dramatically lower in-hive movements and overall poor health, these colonies were still alive after insecticide exposure when behaviours were monitored. Unexpectedly, insecticide use also led to an increased number of honeypots. Bumblebee production of honeypots is a complex response that integrates resource quality with colony-level regulations and feedbacks [49,50]. Given that bumblebees cannot taste neonicotinoids and, in some conditions, prefer imidacloprid-laced nectar [51], the high concentrations in our flowers could have led to preferential nectar (versus pollen) foraging, even though these are complementary rather than substitutable resources. Alternatively, bees may shift to relatively more nectar-foraging as an avoidance response because neonicotinoids tend to occur at much lower concentrations in nectar than pollen [52]. Because we only measured floral visitation, potential shifts in nectar versus pollen collection across treatments are unknown.

Insecticides suppressed watermelon flower visitation by greater than 90%, which ultimately led to lower fruit set. This occurred in spite of using a relatively high stocking rate in our arenas, which contained one hive for a 58 $m^2$ area. For comparison, recommendations for commercial vegetable production are one to three bumblebee hives per 1000 $m^2$ of enclosed space for indoor pollination. This means that bumblebees in our arenas were at least five times higher than the recommended stocking rate for optimal pollination. Furthermore, watermelon requires relatively few visits from bumblebees for successful pollination [25,32,33]. As a result, strong insecticide-mediated reductions in crop foraging appeared to offset the fact that efficient pollinators were present at such high densities.

It is also important to note that clover benefited bumblebee performance compared to a monofloral diet of the crop alone (e.g. increasing live eggs by nearly 400%) without reducing watermelon visitation or fruit set. The average fruit weight was lower when bordered by clover; however, we strongly suspect this is not a pollination-related mechanism but rather a consequence of the unusually close proximity and large size of the clover, resulting in resource competition with the crop (see the picture in the electronic supplementary material, figure S2 for context). This means that non-crop forage as a habitat manipulation can achieve the goal of enhancing pollinator health without detracting from crop pollination by 'pulling' bees away from the crop or clogging stigmas with non-crop pollen. This conclusion is noteworthy and probably extends to more realistic field conditions, given the nature of our experimental set-up. Inside arenas, watermelon and clover were at roughly equal proportions and directly neighbouring one another (i.e. clover was cultivated as a uniform ground cover between each crop row). This is an extreme case compared to a typical agricultural field where the crop/non-crop flower ratio would be far more skewed towards crop dominance, even in diverse systems, and most of the non-crop forage would probably be relegated to field edges rather than growing in such close proximity to the crop itself. Because clover did not interfere with crop pollination under such extreme conditions, it seems highly unlikely in open-field settings that wildflowers would be anything aside from beneficial. More work is needed to understand why clover or other non-crop flowers benefit bees. In this case, it remains unclear whether the beneficial properties derive from higher resource quantity (i.e. twice the number of flowers), quality (i.e. clover pollen is considered high quality with protein levels ranging from 22 to 25%; [53]) or diversity (i.e. one versus several species of pollen). These three features are confounded in our design and thus we cannot disentangle them from one another. However, a recent study using a novel stoichiometric approach to evaluate bee diets based on elemental ratios in pollen, highlighted clover—out of ca 100 taxa considered—as a nutritionally well-balanced species [54]. Cucurbit pollen, on the other hand, is detrimental for B. impatiens performance [39], and, therefore, clover may simply dilute the effects of a suboptimal diet consisting of watermelon alone.

## 5. Conclusion

As a whole, our data indicate that both insecticide use and non-crop flower presence contribute strongly to pollinator health in agroecosystems. An important caveat to this conclusion is that our experimental set-up represents an extreme test of what bees might face under real world conditions. For instance, non-crop flowers (e.g. unmanaged habitats, border vegetation) are typically within the flight range, particularly for large taxa like bumblebees or honeybees, and thus, it would be rare for foragers to be unable to access any floral resource aside from the crop itself. Similarly, insecticide use is usually not an all-or-nothing endeavour; growers vary along a continuum of application frequency and product toxicity. Thus, our experiment should be viewed as a proof-of-concept for the relative importance and interactions among these common stressors, rather than a true estimate of the magnitude of effects experienced in the field.

It is difficult to make broad statements about the relative importance of these factors and the potential for ecological interactions between the two. These conclusions are shaped by the specific aspects of 'health' that are measured or targeted for enhancement. Given that hive-scale survival is critical to bumblebees fulfilling their functional role as crop pollinators, we view the synergistic increase in mortality to be an overriding outcome from this work. The challenge will be to find opportunities for reducing pesticide inputs while maintaining farm productivity. This seems possible both in watermelon [36] and farming systems more generally [55] using IPM and related approaches aimed at increasing sustainability. Achieving this balance is particularly critical in pollinator-dependent crops, like watermelon, where pest management and pollinator conservation are central, but sometimes conflicting, goals.

Finally, it should be noted that, while bumblebee hives were used in this study as a tool for measuring pollinator responses, the broader implications of these findings extend beyond managed species to inform the conservation of wild bees on or near agricultural lands. Managed bumblebee hives are commonly used as convenient proxies for wild taxa. Compared to bumblebees, however, many wild bees tend to have a smaller foraging radius and more limited diet, making the experimental design perhaps even more relevant to these functionally and taxonomically related species. Extending stressor interaction studies to wild bees will be a critical step in understanding the factor(s) contributing to their persistence and diversity in agroecosystems.

Data accessibility. Raw data are available to download via the Purdue University Research Repository at: https://purr.purdue.edu/publications/3754/1 [56].

Authors' contributions. L.L.I.: conceptualization, data curation, investigation, methodology, project administration, writing—original draft, writing—review and editing; J.J.T.: conceptualization, data curation, formal analysis, investigation, methodology, project administration, resources, software, visualization, writing—original draft, writing—review and editing; J.R.P.: investigation, methodology; I.K.: conceptualization, data curation, formal analysis, funding acquisition, investigation, project administration, supervision, visualization, writing—original draft, writing—review and editing. All authors gave final approval for publication and agreed to be held accountable for the work performed therein.

Competing interests. The authors declare we have no competing interests.

Funding. This research was funded by a USDA NIFA grant (no. 2016-51181-25410) to I.K.

Acknowledgements. We thank Amber Jannasch and Yu Han-Hallet from the Purdue Bindley Bioscience Center for their assistance in the LC–MS pesticide residue quantification; Danielle Madison and Natalie Eason for helping with data collection and plot maintenance; and Allison Bistline-East, Wadih Ghanem, Ashley Leach, Christie Shee and Emily Tronson for helpful edits on later drafts of the manuscript.

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
