## [Peer Review File · Proceedings of the Royal Society B: Biological Sciences]

Review History

RSPB-2020-2256.R0 (Original submission)

Review form: Reviewer 1

Recommendation

Major revision is needed (please make suggestions in comments)

Scientific importance: Is the manuscript an original and important contribution to its field?

Good

General interest: Is the paper of sufficient general interest?

Good

Quality of the paper: Is the overall quality of the paper suitable?

Acceptable

Is the length of the paper justified?

Yes

Should the paper be seen by a specialist statistical reviewer?

No

Do you have any concerns about statistical analyses in this paper? If so, please specify them explicitly in your report.

Yes

It is a condition of publication that authors make their supporting data, code and materials available - either as supplementary material or hosted in an external repository. Please rate, if applicable, the supporting data on the following criteria.

Is it accessible?

No

Is it clear?

N/A

Is it adequate?

N/A

Do you have any ethical concerns with this paper?

No

Comments to the Author

Review for manuscript ID: RSPB-2020-2256 Supplemental forage ameliorates the negative impact of insecticides on bumble bees in a pollinator-dependent crop.

This paper investigates the performance of bumblebee colonies pollinating watermelon with/without insecticide treatment and with/without additional non-crop forage. This is an important topic and it's great to see an experiment simultaneously investigating two different pressures on bumblebees under semi-field conditions. However, I have a number of comments that I would like to see addressed/clarified before I recommend the manuscript for publication.

General comments:

Firstly, both the abstract and the introduction start with the difference between the direct and indirect effects of pesticides (from insecticides and herbicides respectively). This study does not look at the impacts of herbicides at all - only the addition of non-crop forage. As this is a different topic, I suggest you remove the references to herbicides as it only adds confusion.

Another significant concern I have is with the statistical analysis, which is currently a little too simplistic to fully interpret the results. The initial colony weight (or initial number of workers) needs to be added into each model as a co-variate, as this is known to have a large effect on the later success of the colony (eg. Whitehorn et al. 2012 *Science* 336 (6079), 351-352). Also, tunnel and time of year should be specified as random effects within each model, to account for any variations these may cause.

Specific comments:

Line 42: Is there evidence that non-crop pollen can act as a toxicity buffer?

Lines 70: This reference would be good to mention here, as it shows that wildflower strips do increase pollinator visits to crops (Feltham, H. et al. 2015. *Experimental evidence that wildflower strips increase pollinator visits to crops. Ecology and evolution*, 5(16), pp.3523-3530).

Lines 150-157: Are these the recommended doses for this crop? So would you expect to find similar concentrations of these pesticides in fields where this crop is grown?

Line 174: Did you only measure the initial and final weight? It would have been good to weigh the colonies weekly to show the initial increase in weight, the peak weight and subsequent decrease. Your measures only show an overall loss (table 2), which does not capture the real picture.

Line 187: 15 female watermelon flowers were tagged to assess pollination efficiency. Is this a big

enough sample to determine this accurately? (I'm not sure how many flower would generally occur in an area of crop this size). Do you have any additional information on the overall harvest for each treatment, or the final size of the fruit?

Line 205: Why do you not later present the results for the total number of queens? This is a useful metric – perhaps more so than the queen weight.

Lines 276-281: Means and standard errors would be more informative here, to get an idea of variation within each treatment group.

Line 280: Actually hive weight loss (shown in Table 2) – perhaps this needs some more explanation.

Line 285: A mean fruit set of 31% seems very low – is this normal for this crop?

Line 287 & Table 3: These pesticide residues are shockingly high for imidacloprid. It's definitely important to know whether the doses you applied here are comparable to the doses farmers use. Also, it is surprising that the clover is so contaminated, even when the watermelon has not been treated. Table 3 shows that the range is up to a higher concentration in clover, in the 'no insecticide treatment'. What is the source of this contamination? On line 297 you state that thiamethoxam & clothianidin were also detected, mostly below the limit of detected – nevertheless, I think it would be good to report this data as well.

Line 321: But it is also detected in clover adjacent to untreated watermelon, this needs to be acknowledged and explained.

Table 3: I am a little confused by a couple of things in this table. Firstly, you said there were only 6 replicates per treatment – why are there two sample sizes of 7 (bumblebee nest material & soil) for a treatment? Also, you are missing a lower value for the range in the Insect. (-) / NCF (+) bumblebee nest material.

Figure 1: These are nice photos but they are not really useful in explaining the experimental design – there is no information about the pesticide treatments here. Is there a way you could present the experimental design clearly and incorporate the photos?

Figure 2: The bottom right panel shows three colonies died on one day – were these all within one time of year block? Was it particularly hot on this day? Also, what is the small cross on the line in the top right panel?

Figure 3: This is a bit tricky to interpret in its current form. Panel A is perhaps not necessary as it is simply the sum of panels B and C. The way you have indicated significance is not clear – brackets connecting the box plots would work better (see figures in Stanley, D. et al 2015. Neonicotinoid pesticide exposure impairs crop pollination services provided by bumblebees. *Nature*, 528(7583), pp.548-550 as an example).

Figure 4: See comment above (significance brackets). Also, a bit more explanation here on the y-axis would be useful (maybe in the legend) – what exactly is 'in-hive activity' in percent?

Review form: Reviewer 2

Recommendation

Major revision is needed (please make suggestions in comments)

Scientific importance: Is the manuscript an original and important contribution to its field?

Excellent

General interest: Is the paper of sufficient general interest?

Excellent

Quality of the paper: Is the overall quality of the paper suitable?

Excellent

Is the length of the paper justified?

Yes

Should the paper be seen by a specialist statistical reviewer?

No

Do you have any concerns about statistical analyses in this paper? If so, please specify them explicitly in your report.

No

It is a condition of publication that authors make their supporting data, code and materials available - either as supplementary material or hosted in an external repository. Please rate, if applicable, the supporting data on the following criteria.

Is it accessible?

No

Is it clear?

N/A

Is it adequate?

N/A

Do you have any ethical concerns with this paper?

No

Comments to the Author

Overview: This study fills an important research gap on the interaction of multiple environmental stressors that bees encounter within agroecosystems. Authors used a clever cage setup to create a two-factorial design manipulating nutrient stress and pesticide exposure. My overall comment is that I think the study implications should be somewhat more restricted, given the limited scope of the system. It's an incredibly important proof of concept for something many within the field assume is the case, or have attempted to demonstrate in the field. However, the setup is an extreme example of what bees might face. It's relatively rare that bees with relatively large foraging ranges, like bumble bees or honey bees, wouldn't be able to access any other floral resources within an agricultural landscape. Weed control isn't perfect, and often there are unmanaged habitats within the flight range of bees in most landscapes. While I don't think this diminishes from the importance of this work, I do think the discussion of the primary results should be put within this more limited scope a bit more. My other comments are relatively minor and can be found below. Though I do think some changes to the figures and tables would be useful to make them easier to interpret.

General comments:

1. Please review and cite the recent Stuligross and Williams *Osmia* paper "Pesticide and resource stressors additively impair wild bee reproduction." *Proceedings B*.

2. Did you count # of reproductives (new queens and males) produced at the end of the study? It might be interesting to explore sex ratios, and potential causes of differences. Though I'm not sure we really understand why some colonies produce more males over new queens. Also, is it possible colonies that are resource constrained might be switching to "end of season" mode quicker? So switching to producing reproductives earlier?

3. In your concluding statements, I'm not sure the implications for IPPM are the strongest takeaway. It's certainly a big deal to show how interacting stressors impact managed bees, but I think the implications for conservation are also incredibly important. Because in all honesty, do we care about the health of Koppert hives once they've "served their purpose?" It could be argued that supplementing the local *B. impatiens* populations with managed colonies isn't the best conservation strategy. But I think it IS important to note that wild colonies nearby

agricultural fields are likely to be experiencing similar stressors, and this really could be having serious conservation consequences. We often use managed colonies as proxies for wild colonies (the pros/cons of this can be debated...) and I think it's important to note the broader conservation implications at the end here.

Line comments:

L166: Did you also remove the pollen patties Koppert typically ships hives with?

L172: I've had issues with weighing colonies outdoors if they are still in their cardboard boxes. Humidity really affects the weight of the cardboard box. Removing the plastic internal box from the cardboard box helps this. Nothing you can do about it now, but might be why you didn't see any effects with weight.

L204: Did you monitor the colonies when you placed them in the freezer? I've found that colonies can survive a couple days in the freezer! And likely are using up stored resources during this time as they try to regulate their temperature :(We've switched to flash freezing them with liquid nitrogen to avoid this issue if we are interested in assessing their stored resources at time of collection.

L263: I think it would be really useful to include some ranges or averages in text as well. Percentages are also useful, but some additional context for how visitation changed would help. Actually, throughout the results section when only percent change is used, I think ranges or averages would be useful as well.

L278: I think more discussion of why bees treated with neonics would have increased nectar stores is needed.

L280: Hard to know how much the hives would increase in weight. You're getting them from Koppert usually at somewhat "peak" production. I've usually found they gain a bit of weight but then tail off pretty quickly as they change over to producing reproductives. Usually if I'm really interested in how colony size changes due to a stressor I will use younger colonies at the start.

L288: This is super interesting. I think more discussion on the fact that you're detecting neonics in all matrices is needed, along with the potential implications for your data.

L295: This is quite a high detection! Was this out of the norm? I think more context for this number would be useful.

L315: "clover presence did not affect the visitation rate on watermelon flowers." This is so important!! And something we hear from growers all the time when we suggest adding "competing" floral resources. Very cool to show that it doesn't impact visitation. I'm curious though, do clover and watermelon bloom/produce nectar at overlapping times during the day? Is it possible they really aren't competing?

Figures/Tables:

Table 1: The directionality of effect is not clear. Can you use the shading for this, instead of how it's currently being used?

Table 2: Might be useful to include some indication of when there was a sig. difference between columns

Figure 2: I'm not sure you really need all 4 panels if you need to save some space. Could just include the "non-crop forage -, insecticide +" one? Or could you combine all into one panel? Just seems to take up a lot of space when only that 4th panel is the one people will be looking at

closely.

Figure 3: I find this figure a bit hard to digest. Could you try it as a stacked barplot with non-crop forage (+/-) stacked on each other? Or maybe just clearer labels "with clover" or "without clover" or something. I think the +/- labels are also adding to some difficulties in comprehension.

Figure 4: I think some more discussion around why in-hive activity is different between treatments would be useful. Might activity be lower when there is no clover because there aren't any resources available? Do watermelon flowers produce nectar all day? Are they open all day? Also, was there really no activity in the insecticide treated, no forage group? Did this include dead hives? Pretty wild. More context for this might be useful.

Decision letter (RSPB-2020-2256.R0)

23-Oct-2020

Dear Dr Kaplan:

I am writing to inform you that your manuscript RSPB-2020-2256 entitled "Supplemental forage ameliorates the negative impact of insecticides on bumble bees in a pollinator-dependent crop" has, in its current form, been rejected for publication in Proceedings B.

This action has been taken on the advice of referees, who have recommended that substantial revisions are necessary. With this in mind we would be happy to consider a resubmission, provided the comments of the referees are fully addressed. However please note that this is not a provisional acceptance.

Sincerely,
 Dr Sasha Dall
 mailto: proceedingsb@royalsociety.org

Associate Editor

Board Member: 1

Comments to Author:

Your manuscript has received two expert and detailed reviews. I have gone back through the manuscript with these reviews in hand. I still believe that your manuscript has significant potential, but if you resubmit then a revision needs to engage with and address all of the points raised by both reviewers, both in the text of the manuscript and the accompanying cover letter. It is certainly possible that re-analyses of your data (as recommended by one reviewer) may change your results and conclusions, which could impact the importance of your manuscript. I won't emphasise any particular aspect of the reviewer comments, as I don't think that they conflict, and I think that all their points are in need of addressing. I hope that you find the reviewer comments helpful, and I look forward to seeing a revised version.

Reviewer(s)' Comments to Author:

Referee: 1

Comments to the Author(s)

Review for manuscript ID: RSPB-2020-2256 Supplemental forage ameliorates the negative impact of insecticides on bumble bees in a pollinator-dependent crop.

This paper investigates the performance of bumblebee colonies pollinating watermelon with/without insecticide treatment and with/without additional non-crop forage. This is an important topic and it's great to see an experiment simultaneously investigating two different pressures on bumblebees under semi-field conditions. However, I have a number of comments that I would like to see addressed/clarified before I recommend the manuscript for publication.

General comments:

Firstly, both the abstract and the introduction start with the difference between the direct and indirect effects of pesticides (from insecticides and herbicides respectively). This study does not look at the impacts of herbicides at all – only the addition of non-crop forage. As this is a different topic, I suggest you remove the references to herbicides as it only adds confusion.

Another significant concern I have is with the statistical analysis, which is currently a little too simplistic to fully interpret the results. The initial colony weight (or initial number of workers) needs to be added into each model as a co-variate, as this is known to have a large effect on the later success of the colony (eg. Whitehorn et al. 2012 *Science* 336 (6079), 351-352). Also, tunnel and time of year should be specified as random effects within each model, to account for any variations these may cause.

Specific comments:

Line 42: Is there evidence that non-crop pollen can act as a toxicity buffer?

Lines 70: This reference would be good to mention here, as it shows that wildflower strips do increase pollinator visits to crops (Feltham, H. et al. 2015. *Experimental evidence that wildflower strips increase pollinator visits to crops. Ecology and evolution*, 5(16), pp.3523-3530).

Lines 150-157: Are these the recommended doses for this crop? So would you expect to find similar concentrations of these pesticides in fields where this crop is grown?

Line 174: Did you only measure the initial and final weight? It would have been good to weigh the colonies weekly to show the initial increase in weight, the peak weight and subsequent decrease. Your measures only show an overall loss (table 2), which does not capture the real picture.

Line 187: 15 female watermelon flowers were tagged to assess pollination efficiency. Is this a big enough sample to determine this accurately? (I'm not sure how many flowers would generally occur in an area of crop this size). Do you have any additional information on the overall harvest for each treatment, or the final size of the fruit?

Line 205: Why do you not later present the results for the total number of queens? This is a useful metric – perhaps more so than the queen weight.

Lines 276-281: Means and standard errors would be more informative here, to get an idea of variation within each treatment group.

Line 280: Actually hive weight loss (shown in Table 2) – perhaps this needs some more explanation.

Line 285: A mean fruit set of 31% seems very low – is this normal for this crop?

Line 287 & Table 3: These pesticide residues are shockingly high for imidacloprid. It's definitely important to know whether the doses you applied here are comparable to the doses farmers use. Also, it is surprising that the clover is so contaminated, even when the watermelon has not been treated. Table 3 shows that the range is up to a higher concentration in clover, in the 'no insecticide treatment'. What is the source of this contamination? On line 297 you state that thiamethoxam & clothianidin were also detected, mostly below the limit of detected – nevertheless, I think it would be good to report this data as well.

Line 321: But it is also detected in clover adjacent to untreated watermelon, this needs to be acknowledged and explained.

Table 3: I am a little confused by a couple of things in this table. Firstly, you said there were only 6 replicates per treatment – why are there two sample sizes of 7 (bumblebee nest material & soil) for a treatment? Also, you are missing a lower value for the range in the Insect. (-) / NCF (+) bumblebee nest material.

Figure 1: These are nice photos but they are not really useful in explaining the experimental design – there is no information about the pesticide treatments here. Is there a way you could present the experimental design clearly and incorporate the photos?

Figure 2: The bottom right panel shows three colonies died on one day – were these all within one time of year block? Was it particularly hot on this day? Also, what is the small cross on the line in the top right panel?

Figure 3: This is a bit tricky to interpret in its current form. Panel A is perhaps not necessary as it is simply the sum of panels B and C. The way you have indicated significance is not clear – brackets connecting the box plots would work better (see figures in Stanley, D. et al 2015. Neonicotinoid pesticide exposure impairs crop pollination services provided by bumblebees. *Nature*, 528(7583), pp.548-550 as an example).

Figure 4: See comment above (significance brackets). Also, a bit more explanation here on the y-axis would be useful (maybe in the legend) – what exactly is 'in-hive activity' in percent?

Referee: 2

Comments to the Author(s)

Overview: This study fills an important research gap on the interaction of multiple environmental stressors that bees encounter within agroecosystems. Authors used a clever cage setup to create a two-factorial design manipulating nutrient stress and pesticide exposure. My overall comment is that I think the study implications should be somewhat more restricted, given the limited scope of the system. It's an incredibly important proof of concept for something many within the field assume is the case, or have attempted to demonstrate in the field. However, the setup is an extreme example of what bees might face. It's relatively rare that bees with relatively large foraging ranges, like bumble bees or honey bees, wouldn't be able to access any other floral resources within an agricultural landscape. Weed control isn't perfect, and often there are unmanaged habitats within the flight range of bees in most landscapes. While I don't think this diminishes from the importance of this work, I do think the discussion of the primary results should be put within this more limited scope a bit more. My other comments are relatively minor and can be found below. Though I do think some changes to the figures and tables would be useful to make them easier to interpret.

General comments:

1. Please review and cite the recent Stuligross and Williams *Osmia* paper "Pesticide and resource stressors additively impair wild bee reproduction." *Proceedings B*.

2. Did you count # of reproductives (new queens and males) produced at the end of the study? It might be interesting to explore sex ratios, and potential causes of differences. Though I'm not sure we really understand why some colonies produce more males over new queens. Also, is it possible colonies that are resource constrained might be switching to "end of season" mode quicker? So switching to producing reproductives earlier?

3. In your concluding statements, I'm not sure the implications for IPPM are the strongest takeaway. It's certainly a big deal to show how interacting stressors impact managed bees, but I think the implications for conservation are also incredibly important. Because in all honesty, do we care about the health of Koppert hives once they've "served their purpose?" It could be argued that supplementing the local *B. impatiens* populations with managed colonies isn't the best conservation strategy. But I think it IS important to note that wild colonies nearby agricultural fields are likely to be experiencing similar stressors, and this really could be having serious conservation consequences. We often use managed colonies as proxies for wild colonies (the pros/cons of this can be debated...) and I think it's important to note the broader conservation implications at the end here.

Line comments:

L166: Did you also remove the pollen patties Koppert typically ships hives with?

L172: I've had issues with weighing colonies outdoors if they are still in their cardboard boxes. Humidity really affects the weight of the cardboard box. Removing the plastic internal box from the cardboard box helps this. Nothing you can do about it now, but might be why you didn't see any effects with weight.

L204: Did you monitor the colonies when you placed them in the freezer? I've found that colonies can survive a couple days in the freezer! And likely are using up stored resources during this time as they try to regulate their temperature :(We've switched to flash freezing them with liquid nitrogen to avoid this issue if we are interested in assessing their stored resources at time of collection.

L263: I think it would be really useful to include some ranges or averages in text as well. Percentages are also useful, but some additional context for how visitation changed would help. Actually, throughout the results section when only percent change is used, I think ranges or averages would be useful as well.

L278: I think more discussion of why bees treated with neonics would have increased nectar stores is needed.

L280: Hard to know how much the hives would increase in weight. You're getting them from Koppert usually at somewhat "peak" production. I've usually found they gain a bit of weight but then tail off pretty quickly as they change over to producing reproductives. Usually if I'm really interested in how colony size changes due to a stressor I will use younger colonies at the start.

L288: This is super interesting. I think more discussion on the fact that you're detecting neonics in all matrices is needed, along with the potential implications for your data.

L295: This is quite a high detection! Was this out of the norm? I think more context for this number would be useful.

L315: "clover presence did not affect the visitation rate on watermelon flowers." This is so important!! And something we hear from growers all the time when we suggest adding "competing" floral resources. Very cool to show that it doesn't impact visitation. I'm curious though, do clover and watermelon bloom/produce nectar at overlapping times during the day? Is it possible they really aren't competing?

Figures/Tables:

Table 1: The directionality of effect is not clear. Can you use the shading for this, instead of how it's currently being used?

Table 2: Might be useful to include some indication of when there was a sig. difference between columns

Figure 2: I'm not sure you really need all 4 panels if you need to save some space. Could just include the "non-crop forage -, insecticide +" one? Or could you combine all into one panel? Just seems to take up a lot of space when only that 4th panel is the one people will be looking at closely.

Figure 3: I find this figure a bit hard to digest. Could you try it as a stacked barplot with non-crop forage (+/-) stacked on each other? Or maybe just clearer labels "with clover" or "without clover" or something. I think the +/- labels are also adding to some difficulties in comprehension.

Figure 4: I think some more discussion around why in-hive activity is different between treatments would be useful. Might activity be lower when there is no clover because there aren't any resources available? Do watermelon flowers produce nectar all day? Are they open all day? Also, was there really no activity in the insecticide treated, no forage group? Did this include dead hives? Pretty wild. More context for this might be useful.

Author's Response to Decision Letter for (RSPB-2020-2256.R0)

See Appendix A.

RSPB-2021-0785.R0

Review form: Reviewer 1

Recommendation

Accept with minor revision (please list in comments)

Scientific importance: Is the manuscript an original and important contribution to its field?

Good

General interest: Is the paper of sufficient general interest?

Excellent

Quality of the paper: Is the overall quality of the paper suitable?

Good

Is the length of the paper justified?

Yes

Should the paper be seen by a specialist statistical reviewer?

No

Do you have any concerns about statistical analyses in this paper? If so, please specify them explicitly in your report.

No

It is a condition of publication that authors make their supporting data, code and materials available - either as supplementary material or hosted in an external repository. Please rate, if applicable, the supporting data on the following criteria.

Is it accessible?

Yes

Is it clear?

Yes

Is it adequate?

Yes

Do you have any ethical concerns with this paper?

No

Comments to the Author

Review for manuscript ID: RSPB-2021-0785 Supplemental forage ameliorates the negative impact of insecticides on bumble bees in a pollinator-dependent crop.

This manuscript fills a knowledge gap, looking at the interaction between forage availability and insecticides in a semi-field experiment. It's a great study and the manuscript has been significantly improved after incorporating the reviewer's comments. My main remaining reservation is about the measurements of 'hive performance', which I think are not very robust - the hives were only weighed at the beginning and end of the experiment, so the important change in weight over a colony's life is not considered (and peak weight is unknown). And the colonies unfortunately produced no new reproductives, so the measurements that were made (number of 'live eggs', 'dead pupae' etc) are not very robust measures of colony fitness. However, the other results on colony survival, flower visitation and crop pollination are very interesting and valuable and overall the manuscript is worthy of publication.

I have a few remaining minor comments:

Firstly, Table 2 and Figure 1 seem to be missing from this version of the manuscript (table 2 referred to in response to comment #9 - or do you mean the supplementary table?).

Line 16: 'are most valued' is slightly misleading. I recommend changing to 'is being considered'.

Line 31: I think there is a typo in this line as 'from adoption of GMO crops' doesn't make sense in this sentence.

Line 177: As the number of queens in each colony was never more than one, I assume that this is the old mother queen? I think you need to clarify that no new queens were produced by the colonies in this experiment, as that is often used as a measure of fitness. Do you have any records of males produced? Also, were there any queen cells filled with nectar? (I assume not, if there were no new queens) - if there were not any, you could remove this metric from the methods.

Line 231, 233 etc: The x bar written by itself is not something I've seen before, I would suggest changing it to something a bit clearer.

Line 264: Individual fruit weights were reduced by clover presence - how about total fruit weight? If there was a greater fruit set in these treatments perhaps the total fruit weight was greater - even if it was made from smaller fruit?

Line 306, 324 and response to comment #13: I think it would be worth mentioning the clover contaminated with imidacloprid from the 'no insecticide treatment' somewhere in the main manuscript (or supplementary materials) as I think it would be something that readers might wonder about.

Line 371 (and line 12): I'm not convinced that an increase in live eggs really shows that clover has been 'a great benefit to bumblebee performance' - especially as it occurs alongside a significantly lower number of worker larvae and worker pupae (from table 1) and comes at the end of the colonies' life. I guess it would be if these were to develop into reproductives, but unfortunately this is unknown. Perhaps it would be good to combine this figure of a higher number of live eggs with the greater queen and worker weight found in these treatments?

Table 1: It would help to clarify in the legend that this table is comparing against the 'null' treatment - so Insect. (-)/NCF (-) (if I am correct with this interpretation!). And I think better to say 'Insecticide + NCF' instead of 'Insecticide x NCF'. Also, there are some significant results that have no colour shading. It would actually be really good to know the direction of all the results, irrespective of significance (particularly as some are close to significance) - would that be possible?

Figures 4 & 5: It would be good show the significance differences (of lack of) between the NCF (-) and NCF (+) bars as well.

Review form: Reviewer 2

Recommendation

Accept as is

Scientific importance: Is the manuscript an original and important contribution to its field?

Excellent

General interest: Is the paper of sufficient general interest?

Excellent

Quality of the paper: Is the overall quality of the paper suitable?

Excellent

Is the length of the paper justified?

Yes

Should the paper be seen by a specialist statistical reviewer?

No

Do you have any concerns about statistical analyses in this paper? If so, please specify them explicitly in your report.

No

It is a condition of publication that authors make their supporting data, code and materials available - either as supplementary material or hosted in an external repository. Please rate, if applicable, the supporting data on the following criteria.

Is it accessible?

Yes

Is it clear?

Yes

Is it adequate?

Yes

Do you have any ethical concerns with this paper?

No

Comments to the Author

The authors did a wonderful job incorporating reviewer comments. I think this is a great article, and I don't have any further comments/edits. Well done!

Decision letter (RSPB-2021-0785.R0)

17-May-2021

Dear Dr Kaplan:

Your manuscript has now been peer reviewed and the reviews have been assessed by an Associate Editor. The reviewers' comments (not including confidential comments to the Editor) and the comments from the Associate Editor are included at the end of this email for your reference. As you will see, the reviewers and the Editors have raised some concerns with your manuscript and we would like to invite you to revise your manuscript to address them.

Research ethics:

Use of animals and field studies:

It is a condition of publication that you make available the data and research materials supporting the results in the article (<https://royalsociety.org/journals/authors/author-guidelines/#data>). Datasets should be deposited in an appropriate publicly available repository and details of the associated accession number, link or DOI to the datasets must be included in the Data Accessibility section of the article (<https://royalsociety.org/journals/ethics-policies/data-sharing-mining/>). Reference(s) to datasets should also be included in the reference list of the article with DOIs (where available).

Please submit a copy of your revised paper within three weeks. If we do not hear from you within this time your manuscript will be rejected. If you are unable to meet this deadline please let us know as soon as possible, as we may be able to grant a short extension.

Best wishes,

Dr Sasha Dall
 mailto: proceedingsb@royalsociety.org

Associate Editor
 Comments to Author:

Your manuscript has been reviewed again by both reviewers. They, like I, are happy with your revisions, but there remain a number of issues that need to be addressed. One reviewer makes a series of recommendations and comments, and these all need to be addressed in a cover letter and through revisions in a revised version of the manuscript (I have gone through these comments and they all require changes in the manuscript, so please don't simply address them in a cover letter). I have made some additional points below, which also need to be addressed. If you are prepared to submit a revised manuscript, please make sure to upload a tracked changes version so I can see exactly how and where you've made changes.

[line numbers as per original word document submission]

line 31 - change 'from' to 'due to'

lines 97-98 - using 'even fewer' here makes no sense in the way the sentence is written. As you are using bumblebees in your experiment, the focus on honey bee visits in this sentence also makes no sense. Please rewrite to put the focus on the number of bumblebee visits (and give their quantity, not that for honey bees).

lines 149-150 - it is not clear what metric you used for 'survival' - please add specific information of what you measured here (was it, for example, all workers plus mother queen were dead?)

line 175 - how did you measure worker weight? was it measured for all workers and then averaged for the colony for analysis? please clarify this metric. If it's not one value per hive, then hive should be included as a random factor in your analysis - again, please clarify

line 178 - how did you measure 'live eggs'? Please define how you knew eggs were alive here. If you didn't actually measure life, but simply measured the presence of eggs, then rewrite to make this clear (and then change language in other sections of the paper appropriately, if necessary)

line 264 - 'Alternatively' is not the right word here. Try 'In contrast' or something like that

lines 324-340 - you need to discuss the fact that you found imidacloprid in your 'no insecticide' treatments, and the implications of this for your results

Reviewer(s)' Comments to Author:

Referee: 1

Comments to the Author(s).

Review for manuscript ID: RSPB-2021-0785 Supplemental forage ameliorates the negative impact of insecticides on bumble bees in a pollinator-dependent crop.

This manuscript fills a knowledge gap, looking at the interaction between forage availability and insecticides in a semi-field experiment. It's a great study and the manuscript has been significantly improved after incorporating the reviewer's comments. My main remaining reservation is about the measurements of 'hive performance', which I think are not very robust - the hives were only weighed at the beginning and end of the experiment, so the important change in weight over a colony's life is not considered (and peak weight is unknown). And the colonies unfortunately produced no new reproductives, so the measurements that were made (number of 'live eggs', 'dead pupae' etc) are not very robust measures of colony fitness. However, the other results on colony survival, flower visitation and crop pollination are very interesting and valuable and overall the manuscript is worthy of publication.

I have a few remaining minor comments:

Firstly, Table 2 and Figure 1 seem to be missing from this version of the manuscript (table 2 referred to in response to comment #9 - or do you mean the supplementary table?).

Line 16: 'are most valued' is slightly misleading. I recommend changing to 'is being considered'.

Line 31: I think there is a typo in this line as 'from adoption of GMO crops' doesn't make sense in this sentence.

Line 177: As the number of queens in each colony was never more than one, I assume that this is the old mother queen? I think you need to clarify that no new queens were produced by the colonies in this experiment, as that is often used as a measure of fitness. Do you have any records of males produced? Also, were there any queen cells filled with nectar? (I assume not, if there were no new queens) – if there were not any, you could remove this metric from the methods.

Line 231, 233 etc: The x bar written by itself is not something I've seen before, I would suggest changing it to something a bit clearer.

Line 264: Individual fruit weights were reduced by clover presence - how about total fruit weight? If there was a greater fruit set in these treatments perhaps the total fruit weight was greater – even if it was made from smaller fruit?

Line 306, 324 and response to comment #13: I think it would be worth mentioning the clover contaminated with imidacloprid from the 'no insecticide treatment' somewhere in the main manuscript (or supplementary materials) as I think it would be something that readers might wonder about.

Line 371 (and line 12): I'm not convinced that an increase in live eggs really shows that clover has been 'a great benefit to bumblebee performance' – especially as it occurs alongside a significantly lower number of worker larvae and worker pupae (from table 1) and comes at the end of the colonies' life. I guess it would be if these were to develop into reproductives, but unfortunately this is unknown. Perhaps it would be good to combine this figure of a higher number of live eggs with the greater queen and worker weight found in these treatments?

Table 1: It would help to clarify in the legend that this table is comparing against the 'null' treatment – so Insect. (-)/NCF (-) (if I am correct with this interpretation!). And I think better to say 'Insecticide + NCF' instead of 'Insecticide x NCF'. Also, there are some significant results that have no colour shading. It would actually be really good to know the direction of all the results, irrespective of significance (particularly as some are close to significance) – would that be possible?

Figures 4 & 5: It would be good show the significance differences (of lack of) between the NCF (-) and NCF (+) bars as well.

Referee: 2

Comments to the Author(s).

The authors did a wonderful job incorporating reviewer comments. I think this is a great article, and I don't have any further comments/edits. Well done!

Author's Response to Decision Letter for (RSPB-2021-0785.R0)

See Appendix B.

Decision letter (RSPB-2021-0785.R1)

04-Jun-2021

Dear Dr Kaplan

I am pleased to inform you that your manuscript entitled "Supplemental forage ameliorates the negative impact of insecticides on bumble bees in a pollinator-dependent crop" has been accepted for publication in Proceedings B.

Data Accessibility section

Open Access

Paper charges

Sincerely,

Dr Sasha Dall

Associate Editor:

Board Member

Comments to Author:

Thanks for engaging so thoroughly with the reviewer (and my) comments. The manuscript is clearer as a result, and will therefore be more valuable to readers.

Appendix A

Thank you for the opportunity to revise and resubmit Manuscript ID: RSPB-2020-2256: “*Supplemental forage ameliorates the negative impact of insecticides on bumble bees in a pollinator-dependent crop*”. We apologize for the unusually large delay in getting this revision prepared. Fully addressing all reviewer comments during the COVID-19 pandemic while balancing other professional responsibilities proved challenging! We have now addressed all of the reviewers’ comments, which are detailed below with our responses in bold red font.

Referee: 1

Comment #1: Firstly, both the abstract and the introduction start with the difference between the direct and indirect effects of pesticides (from insecticides and herbicides respectively). This study does not look at the impacts of herbicides at all – only the addition of non-crop forage. As this is a different topic, I suggest you remove the references to herbicides as it only adds confusion.

Response: Thank you for the suggestion. We had a long discussion about this exact issue on the final draft of the original manuscript. Referee 1 is correct that we never specifically test herbicides. To address this, we rewrote the Abstract and Introduction beginnings, removing all reference to direct/indirect effects from pesticides and, instead, focus on insecticides and lack of forage as primary stressors for bees. Herbicides are mentioned briefly once, but only in a supporting role as one of several potential mechanisms causing reductions to the non-crop flowering plant community.

Comment #2: Another significant concern I have is with the statistical analysis, which is currently a little too simplistic to fully interpret the results. The initial colony weight (or initial number of workers) needs to be added into each model as a co-variate, as this is known to have a large effect on the later success of the colony (eg. Whitehorn et al. 2012 Science 336 (6079), 351-352). Also, tunnel and time of year should be specified as random effects within each model, to account for any variations these may cause.

Response: These are all good points. We consulted with a statistician and reanalyzed the data as suggested and, in general, included far more detail in the statistical description. We now include initial colony weight as a co-variate for all hive performance data. We also added time of year (i.e., trial) as an effect in the model; however, since there were only two levels, we modeled this as a fixed effect since random effects designation is only recommended with >5 levels (Bolker et al. 2008). We included spatial block as a factor, which constituted two paired, neighboring tunnels, or one full replicate of the experiment (i.e., tunnels 1/2, 3/4, 5/6).

Bolker BM et al. 2008. Generalized linear mixed models: a practical guide for ecology and evolution. Trends in Ecology & Evolution 24:127-135.

Comment #3: Line 42: Is there evidence that non-crop pollen can act as a toxicity buffer?

Response: There is not but this would be hard to prove. We viewed this more as a general toxicological principle whereby swapping out a more toxic food with a less toxic one will act as a toxicity buffer. We don’t consider this statement too controversial so left as is.

Comment #4: Lines 70: This reference would be good to mention here, as it shows that wildflower strips do increase pollinator visits to crops (Feltham, H. et al. 2015. Experimental

evidence that wildflower strips increase pollinator visits to crops. Ecology and evolution, 5(16), pp.3523-3530).

Response: Thank you for the suggestion. We added the noted citation at this sentence and also included the following recent meta-analysis on the topic:

Zamorano J, Bartomeus I, Grez AA, Garibaldi LA. 2020. Field margin floral enhancements increase pollinator diversity at the field edge but show no consistent spillover into the crop field: a meta- analysis. Insect Conservation & Diversity, in press (doi:10.1111/icad.12454)

Comment #5: Lines 150-157: Are these the recommended doses for this crop? So would you expect to find similar concentrations of these pesticides in fields where this crop is grown?

Response: Sorry, we forgot to include this important information. Yes, we used the dose that commercial growers use. For Admire Pro, the recommended rates listed on the label are a range from 17.3 to 26 oz/ha. In the experiment, we used 24.7 oz/ha. This information was added to the paper in the noted section of the Methods (L125-127 in revision).

Comment #6: Line 174: Did you only measure the initial and final weight? It would have been good to weigh the colonies weekly to show the initial increase in weight, the peak weight and subsequent decrease. Your measures only show an overall loss (table 2), which does not capture the real picture.

Response: Agreed. Weekly measurements would have been better, but we only have beginning and end weights, unfortunately.

Comment #7: Line 187: 15 female watermelon flowers were tagged to assess pollination efficiency. Is this a big enough sample to determine this accurately? (I'm not sure how many flowers would generally occur in an area of crop this size). Do you have any additional information on the overall harvest for each treatment, or the final size of the fruit?

Response: Across both trials, we had 30 flowers tagged per treatment combination, resulting in 180 flowers to test main effects of either insecticides or non-crop forage. This is plenty, in our opinion, for an accurate estimate of pollination efficiency. Although there were numerous flowers in these arenas, watermelon floral sex ratios are highly male-biased (ca. 10:1 male:female flowers) so this limits sample size since we were only interested in female flowers.

We do have data on fruit size from the tagged flowers (i.e., not from the whole tunnel). We did not include this information in the original submission for a few reasons. One, we only have these data for trial 2, not trial 1. Two, clover-arenas produced considerably smaller melons. We suspect this is due to crowding since the two plants were grown in such close proximity. Because this was unrelated to pollination we chose to leave the data out of the original manuscript submission; however, we added these new data into the revision as requested.

Comment #8: Line 205: Why do you not later present the results for the total number of queens? This is a useful metric – perhaps more so than the queen weight.

Response: We left these data out because queen numbers were so low; always either zero or one per hive. This information is now noted in the Results on L255-256.

Comment #9: Lines 276-281: Means and standard errors would be more informative here, to get an idea of variation within each treatment group.

Response: We removed percent change from this passage and now report means for the main effects that were significant. We did not report SEs since these are already include in Table 2.

Comment #10: Line 280: Actually hive weight loss (shown in Table 2) – perhaps this needs some more explanation.

Response: Good point. To simplify, we just removed the directionality of gain/loss and report as “hive weight”.

Comment #11: Line 285: A mean fruit set of 31% seems very low – is this normal for this crop?

Response: Yes, this is right in the range we tend to see with watermelon, which overproduce flowers relative to how many fruits a vine can support. We added a citation for the below paper (L260-261) that has a table listing watermelon % fruit set under a range of pollinator visitations. Values are always <50% and mostly between 20% and 40%, even with hand pollination.

Adlerz WC. 1966. Honey bee visit numbers and watermelon pollination. *Journal of Economic Entomology* 59, 28-30.

Comment #12: Line 287 & Table 3: These pesticide residues are shockingly high for imidacloprid. It’s definitely important to know whether the doses you applied here are comparable to the doses farmers use.

Response: Yep, see response to Comment #5 above. We also included more discussion of how these residues compare with published values for imidacloprid; see response to Comment #33 below for Referee 2.

Comment #13: Also, it is surprising that the clover is so contaminated, even when the watermelon has not been treated. Table 3 shows that the range is up to a higher concentration in clover, in the ‘no insecticide treatment’. What is the source of this contamination?

Response: Referee 1 is absolutely correct that there is one clover sample in the insecticide-free tunnel that had extremely high imidacloprid residues, shown by the max. value in the range. We strongly suspect this was due to contamination, either during the collection or analytical stages; however, there is no way of proving that this is not biologically “real” so we left the data as is (i.e., we did not omit that data point). This is not uncommon in toxicological studies involving pesticides, which is why mean values tend to be far less informative than medians due to the influential effects of one or two large values. To this point, the median value for imidacloprid residues in clover was >50-times higher in insecticide-treated (17.43 ng/g) vs. untreated arenas (0.33 ng/g). Thus, we are confident that bees in the untreated arenas were not inadvertently exposed to high levels of imidacloprid.

Comment #14: On line 297 you state that thiamethoxam & clothianidin were also detected, mostly below the limit of detected – nevertheless, I think it would be good to report this data as well.

Response: Agreed. We added these data at the noted location.

Comment #15: Line 321: But it is also detected in clover adjacent to untreated watermelon, this needs to be acknowledged and explained.

Response: See above response to Comment #13.

Comment #16: Table 3: I am a little confused by a couple of things in this table. Firstly, you said there were only 6 replicates per treatment – why are there two sample sizes of 7 (bumblebee nest material & soil) for a treatment?

Response: Correct, there are 6 replicates. We ended up taking two measurements in one of the tunnels. These two values were averaged to provide a single value per tunnel in the revision.

Comment #17: Also, you are missing a lower value for the range in the Insect. (-) / NCF (+) bumblebee nest material.

Response: This is because there was only a single sample where we detected imidacloprid for this treatment.

Comment #18: Figure 1: These are nice photos but they are not really useful in explaining the experimental design – there is no information about the pesticide treatments here. Is there a way you could present the experimental design clearly and incorporate the photos?

Response: Good point. We remade Fig 1, using it to describe the experimental design. In doing so, we removed the in-hive picture since this was less relevant.

Comment #19: Figure 2: The bottom right panel shows three colonies died on one day – were these all within one time of year block? Was it particularly hot on this day? Also, what is the small cross on the line in the top right panel?

Response: Yes, several hives died in early September (9/5). We went back and checked temperature data in tunnels during this period. The days immediately preceding this date (Sept 1-5) had the highest daily maximum temperatures of the experiment (avg. 106.1°F). The two weeks prior to this (8/16 - 8/31) had maximum temperatures that were >10 degrees cooler (avg. 95.2°F). So, yes, this very much supports the hypothesis that heat stress exacerbated the insecticide/forage synergy by adding a third stressor. This new information was added to the Discussion (L311-316).

The small cross on the line is to indicate censored data, i.e., those removed for factors unrelated to treatments. In this case, we had one hive that was knocked over from high winds and had to be excluded. This cross was removed in the new figure version.

Comment #20: Figure 3: This is a bit tricky to interpret in its current form. Panel A is perhaps not necessary as it is simply the sum of panels B and C. The way you have indicated significance is not clear – brackets connecting the box plots would work better (see figures in Stanley, D. et al 2015. Neonicotinoid pesticide exposure impairs crop pollination services provided by bumblebees. Nature, 528(7583), pp.548-550 as an example).

Response: Yes, we agree. We totally remade Fig 3. In doing so, we removed panel A (summed data) so it's now a two-panel figure with watermelon and clover stacked in A/B panels. In the original figures we used boxes with asterisks to denote significance of main effects for both factors (unlike Stanley et al 2015, which was a one-way ANOVA with 3 treatments). Since there were no statistical interactions between insecticide use and clover

presence, pairwise comparisons among all four treatment groups is not recommended. However, we agree that brackets connecting box plots is more visually appealing and intuitive so we modified the figure to use brackets with a line over paired bars for illustrating significance of the main effect of insecticide use (+/-).

Comment #21: Figure 4: See comment above (significance brackets). Also, a bit more explanation here on the y-axis would be useful (maybe in the legend) – what exactly is ‘in-hive activity’ in percent?

Response: **Yep, we changed this figure as well, like Figure 3. We also added a few sentences to the legend describing more about the response variable.**

Referee: 2

Comment #22: This study fills an important research gap on the interaction of multiple environmental stressors that bees encounter within agroecosystems. Authors used a clever cage setup to create a two-factorial design manipulating nutrient stress and pesticide exposure. My overall comment is that I think the study implications should be somewhat more restricted, given the limited scope of the system. It’s an incredibly important proof of concept for something many within the field assume is the case, or have attempted to demonstrate in the field. However, the setup is an extreme example of what bees might face. It’s relatively rare that bees with relatively large foraging ranges, like bumble bees or honey bees, wouldn’t be able to access any other floral resources within an agricultural landscape. Weed control isn’t perfect, and often there are unmanaged habitats within the flight range of bees in most landscapes. While I don’t think this diminishes from the importance of this work, I do think the discussion of the primary results should be put within this more limited scope a bit more. My other comments are relatively minor and can be found below. Though I do think some changes to the figures and tables would be useful to make them easier to interpret.

Response: **Thanks very much for this thoughtful perspective. We completely agree and rewrote parts of the Discussion to place the experimental design in better context relative to real-world conditions. See lines 398-406.**

Comment #23: Please review and cite the recent Stuligross and Williams *Osmia* paper “Pesticide and resource stressors additively impair wild bee reproduction.” Proceedings B.

Response: **Thanks for pointing this out. Since this study was published after we submitted the original manuscript, we were unable to include it at first but had seen the paper and intended to add it during revisions. It is now cited in the Introduction and Discussion sections.**

Comment #24: Did you count # of reproductives (new queens and males) produced at the end of the study? It might be interesting to explore sex ratios, and potential causes of differences. Though I’m not sure we really understand why some colonies produce more males over new queens. Also, is it possible colonies that are resource constrained might be switching to “end of season” mode quicker? So switching to producing reproductives earlier?

Response: **This would be nice to include, but new queen and male numbers were extremely low in dissected hives (always either 0 or 1) so we are unable to test their responses,**

unfortunately. The information on low reproductive counts were added to the Results section (L255-256) in case others wonder about this.

Comment #25: In your concluding statements, I'm not sure the implications for IPPM are the strongest takeaway. It's certainly a big deal to show how interacting stressors impact managed bees, but I think the implications for conservation are also incredibly important. Because in all honesty, do we care about the health of Koppert hives once they've "served their purpose?" It could be argued that supplementing the local *B. impatiens* populations with managed colonies isn't the best conservation strategy. But I think it IS important to note that wild colonies nearby agricultural fields are likely to be experiencing similar stressors, and this really could be having serious conservation consequences. We often use managed colonies as proxies for wild colonies (the pros/cons of this can be debated...) and I think it's important to note the broader conservation implications at the end here.

Response: Absolutely. We agree with everything written in this comment and rewrote the Conclusions section to deemphasize bumble bees specifically, highlighting their use as proxies for wild species and its implications more broadly (see L419-426).

Comment #26: L166: Did you also remove the pollen patties Koppert typically ships hives with?

Response: We ordered research hives that did not include pollen patties. This is now noted in the Methods section (L138).

Comment #27: L172: I've had issues with weighing colonies outdoors if they are still in their cardboard boxes. Humidity really affects the weight of the cardboard box. Removing the plastic internal box from the cardboard box helps this. Nothing you can do about it now, but might be why you didn't see any effects with weight.

Response: We do have weights with just the plastic internal box, removing the cardboard. The problem is that we did not collect these data for initial weight in trial 1 (we have final with trial 1 and initial/final with trial 2), which is why we just stuck with cardboard weights for consistency since we have those data for all time points and trials. However, we went back and looked at data for just trial 2. Mean initial/final weights across all hives, irrespective of treatment, were 3.54 vs. 3.50 kg, respectively. So they still lost a bit of weight over time.

Comment #28: L204: Did you monitor the colonies when you placed them in the freezer? I've found that colonies can survive a couple days in the freezer! And likely are using up stored resources during this time as they try to regulate their temperature :(We've switched to flash freezing them with liquid nitrogen to avoid this issue if we are interested in assessing their stored resources at time of collection.

Response: Thanks for this tip! We did not monitor colonies in the freezer, but this is good to know for future experiments.

Comment #29: L263: I think it would be really useful to include some ranges or averages in text as well. Percentages are also useful, but some additional context for how visitation changed would help. Actually, throughout the results section when only percent change is used, I think ranges or averages would be useful as well.

Response: Yep, we added this information throughout the Results section.

Comment #30: L278: I think more discussion of why bees treated with neonics would have increased nectar stores is needed.

Response: Agreed. This was the most unexpected outcome from the hive data and one unaddressed in the previous manuscript version. We added a new passage in the Discussion on L348-356.

Comment #31: L280: Hard to know how much the hives would increase in weight. You're getting them from Koppert usually at somewhat "peak" production. I've usually found they gain a bit of weight but then tail off pretty quickly as they change over to producing reproductives. Usually if I'm really interested in how colony size changes due to a stressor I will use younger colonies at the start.

Response: We ordered research colonies, which contain fewer workers (70) than hives destined for commercial fields so the colonies should've been a bit younger than normal. We suspect that the lack of hive weight gain observed was due to the stressful conditions we placed them under, i.e., relatively few floral resources (both floral quantity and diversity) and higher temperatures compared to what they would experience in an open-field setting.

Comment #32: L288: This is super interesting. I think more discussion on the fact that you're detecting neonics in all matrices is needed, along with the potential implications for your data.

Response: We added a new paragraph to the Discussion (L322-338) that addresses the routes of exposure issue noted here, as well as pesticide residue concentrations noted in Comment #33 below.

Comment #33: L295: This is quite a high detection! Was this out of the norm? I think more context for this number would be useful.

Response: We agree that detection levels were high. However, it is hard to say if these values are out of the norm since there are no published field data on watermelon and neonicotinoids, only related cucurbits like squash that are in different genera. Nevertheless, we added more comparison of these published values. Also, covered high tunnels are different than open-field environments; they're protected structures that alter inputs and could result in higher residues. For example, there is no rain flushing pesticides from the soil. Also, the plastic covering filters the sunlight potentially leading to altered pesticide degradation rates. These factors are now included in the Discussion in a new paragraph (L322-338).

Comment #34: L315: "clover presence did not affect the visitation rate on watermelon flowers." This is so important!! And something we hear from growers all the time when we suggest adding "competing" floral resources. Very cool to show that it doesn't impact visitation. I'm curious though, do clover and watermelon bloom/produce nectar at overlapping times during the day? Is it possible they really aren't competing?

Response: This is a good question. It is certainly possible that you could create a companion planting system where bloom times over the daily cycle are designed to complement rather than compete with the crop; although we are unaware of such as a study. In this case, watermelon flowers are open from early in the morning (ca. 6 am) to

early afternoon (ca. 2 pm). This coincides with clover availability. It also reflects behaviors noted in tunnels where we spotted bees simultaneously foraging from both clover and watermelon during the same observation time periods. This point is now emphasized in the Discussion on L300-301.

Comment #35: Table 1: The directionality of effect is not clear. Can you use the shading for this, instead of how it's currently being used?

Response: Yes, we changed the shading design so that it now describes directionality rather than significance.

Comment #36: Table 2: Might be useful to include some indication of when there was a sig. difference between columns

Response: Yes, we added asterisks throughout to note the variables with significant main or interactive effects from the treatments.

Comment #37: Figure 2: I'm not sure you really need all 4 panels if you need to save some space. Could just include the "non-crop forage -, insecticide +" one? Or could you combine all into one panel? Just seems to take up a lot of space when only that 4th panel is the one people will be looking at closely.

Response: Agreed. We remade this figure with all four combined in one. Originally, we wanted to avoid this since the three 100% survival curves would be totally overlapping. However, we jittered the lines a tad so they're each visible. We think this new version is way better and takes up far less space. Thanks!

Comment #38: Figure 3: I find this figure a bit hard to digest. Could you try it as a stacked barplot with non-crop forage (+/-) stacked on each other? Or maybe just clearer labels "with clover" or "without clover" or something. I think the +/- labels are also adding to some difficulties in comprehension.

Response: Sorry about this. We totally remade this figure from scratch, stacking the panels, which reduced x/y-axis title redundancy, and using icons to denote watermelon vs. clover response variables. We also removed the A-panel, as recommended by Referee 1, which should make it easier to digest. Regarding the forage treatment, we think stacking the bars would make the four treatments harder to decipher and compare with one another so we kept them separate. We would also prefer to keep +/- since it fits the overall scheme used for insecticides on the x-axis. We have used +/- to denote presence/absence in numerous other publications in the past without any complaints or confusion, so would prefer to keep this notation. Similarly, we prefer to keep NCF since the acronym is used throughout the paper and would like to remain consistent in our treatment labeling. We feel that the other changes we made to the figure (noted above) should have cleared things up for readers but are open to further suggestions.

Comment #39: Figure 4: I think some more discussion around why in-hive activity is different between treatments would be useful. Might activity be lower when there is no clover because there aren't any resources available? Do watermelon flowers produce nectar all day? Are they open all day? Also, was there really no activity in the insecticide treated, no forage group? Did this include dead hives? Pretty wild. More context for this might be useful.

Response: Hive activity did not respond to clover presence/absence so resource availability was not a factor we considered in interpreting these data. However, we added further context for the insecticide effect in the Discussion (L344-347).

Appendix B

Thank you for the opportunity to revise and resubmit Manuscript ID: RSPB-2020-2256: “*Supplemental forage ameliorates the negative impact of insecticides on bumble bees in a pollinator-dependent crop*”. We have now addressed all of the reviewers’ comments, which are detailed below with our responses in bold red font.

Associate Editor

Comment #1: “Please make sure to upload a tracked changes version so I can see exactly how and where you’ve made changes.”

Response: Done. We have both documents up there.

“line 31 - change ‘from’ to ‘due to’”

Response: Done

“lines 97-98 - using ‘even fewer’ here makes no sense in the way the sentence is written. As you are using bumblebees in your experiment, the focus on honey bee visits in this sentence also makes no sense. Please rewrite to put the focus on the number of bumblebee visits (and give their quantity, not that for honey bees).”

Response: Good point. We removed the reference to honey bee visits here and added specifics for bumble bees instead.

“lines 149-150 - it is not clear what metric you used for ‘survival’ - please add specific information of what you measured here (was it, for example, all workers plus mother queen were dead?)”

Response: Done – the following sentence was added to the noted section.

“They were considered dead when no active bees were observed tending the colony or out foraging on flowers for two consecutive surveys.”

“line 175 - how did you measure worker weight? was it measured for all workers and then averaged for the colony for analysis? please clarify this metric. If it’s not one value per hive, then hive should be included as a random factor in your analysis - again, please clarify”

Response: We added the following sentence, which should clarify:

“Weights were calculated by combining all bees from each group (worker or queen), measuring total weight and dividing by the number of individuals, resulting in a single average value per hive.”

“line 178 - how did you measure ‘live eggs’? Please define how you knew eggs were alive here. If you didn’t actually measure life, but simply measured the presence of eggs, then rewrite to make this clear (and then change language in other sections of the paper appropriately, if necessary)”

Response: The language was correct (live vs. dead) so we left this part but included more methodological information in the noted spot on how we differentiated the two. We added the following few sentences:

“Eggs were considered dead when they were desiccated and/or black upon visual inspection. Viable eggs are milky in color, oval-shaped and free of secondary pathogens.”

“line 264 – ‘Alternatively’ is not the right word here. Try ‘In contrast’ or something like that”

Response: Done – changed to “In contrast”.

“lines 324-340 - you need to discuss the fact that you found imidacloprid in your 'no insecticide' treatments, and the implications of this for your results”

Response: Sorry, this was mentioned by one of the reviewers in the prior revision and, although we gave our interpretation of this effect in the cover letter, we did not add it to the manuscript since we did not think the amounts were biologically relevant. However, this was now added to the noted paragraph of the Discussion on L348-356.

Referee: 1

This manuscript fills a knowledge gap, looking at the interaction between forage availability and insecticides in a semi-field experiment. It's a great study and the manuscript has been significantly improved after incorporating the reviewer's comments. My main remaining reservation is about the measurements of 'hive performance', which I think are not very robust – the hives were only weighed at the beginning and end of the experiment, so the important change in weight over a colony's life is not considered (and peak weight is unknown). And the colonies unfortunately produced no new reproductives, so the measurements that were made (number of 'live eggs', 'dead pupae' etc) are not very robust measures of colony fitness. However, the other results on colony survival, flower visitation and crop pollination are very interesting and valuable and overall the manuscript is worthy of publication.

Response: Thank you for the feedback. We agree that the lack of reproductives was unfortunate and weekly weights would have been better than one final weight, but we are encouraged that the reviewer appreciated the other response variables and the study overall!

“Firstly, Table 2 and Figure 1 seem to be missing from this version of the manuscript (table 2 referred to in response to comment #9 – or do you mean the supplementary table?).”

Response: Sorry for not mentioning this in the previous letter since it is confusing. This was indeed changed, but it was not a response to editor or reviewer comments. Unfortunately, we were unable to resubmit the manuscript unless it fit under 10 printed pages. This required some non-trivial changes, which included moving figures, tables, and some text to the supplementary materials. The problem mostly stemmed from adding material during the revision process, which caused the manuscript to increase in size compared to the original submission. None of the original materials were removed from the paper, they were simply converted to online supplementary files.

“Line 16: ‘are most valued’ is slightly misleading. I recommend changing to ‘is being considered’.”

Response: Yes, we like this better too – it was changed to “is being considered”. Thanks for the suggestion.

“Line 31: I think there is a typo in this line as ‘from adoption of GMO crops’ doesn't make sense in this sentence.”

Response: This was fixed in response to editor comment above. We also added “herbicide-tolerant” before “GMO crops” to make this more clear.

“Line 177: As the number of queens in each colony was never more than one, I assume that this is the old mother queen? I think you need to clarify that no new queens were produced by the colonies in this experiment, as that is often used as a measure of fitness. Do you have any records of males produced? Also, were there any queen cells filled with nectar? (I assume not, if there were no new queens) – if there were not any, you could remove this metric from the methods.”

Response: We added the following language in the Results, which should clarify these issues (L260-263):

“Data on number of reproductives (queens and males) were excluded since they were always either zero or one per hive. No new queens were produced by the colonies in this experiment (i.e., in cases where one was found it was likely the old mother queen originating with the hive).”

No, we do not have data on queen cells filled with nectar so this variable was deleted in the Methods, as suggested.

“Line 231, 233 etc: The x bar written by itself is not something I’ve seen before, I would suggest changing it to something a bit clearer.”

Response: We changed this to the word “mean” throughout the Results section.

“Line 264: Individual fruit weights were reduced by clover presence - how about total fruit weight? If there was a greater fruit set in these treatments perhaps the total fruit weight was greater – even if it was made from smaller fruit?”

Response: While we agree with this hypothetical scenario, this would not be possible given that fruit set was unaffected by clover presence and trended to being slightly lower, not higher (see Fig. 4). We added the following sentence at the end of this paragraph:

“Because fruit set was not significantly different, and trended towards being slightly lower, in the presence of clover, changes in this variable could not compensate for the decline in individual fruit weight to affect total fruit weight per arena.”

“Line 306, 324 and response to comment #13: I think it would be worth mentioning the clover contaminated with imidacloprid from the ‘no insecticide treatment’ somewhere in the main manuscript (or supplementary materials) as I think it would be something that readers might wonder about.”

Response: Yes, sorry about not adding this to the prior revision. See response to editor above.

“Line 371 (and line 12): I’m not convinced that an increase in live eggs really shows that clover has been ‘a great benefit to bumblebee performance’ – especially as it occurs alongside a significantly lower number of worker larvae and worker pupae (from table 1) and comes at the end of the colonies’ life. I guess it would be if these were to develop into reproductives, but unfortunately this is unknown. Perhaps it would be good to combine this figure of a higher number of live eggs with the greater queen and worker weight found in these treatments?”

Response: We are not entirely sure what the reviewer is suggesting here. We toned down the language in the noted passage here (removing ‘greatly’) to account for the first part of their comment, but these data are all presented in table form so there is no figure to

combine and we are unsure how they envision combining egg numbers and body weights since they are totally different variables.

“Table 1: It would help to clarify in the legend that this table is comparing against the ‘null’ treatment – so Insect. (-)/NCF (-) (if I am correct with this interpretation!). And I think better to say ‘Insecticide + NCF’ instead of ‘Insecticide x NCF’.”

Response: What we presented here is a standard two-way ANOVA-style statistical table showing the main and interactive effects of insecticide and non-crop forage. This is not comparing back against the null treatment (control), and the “x” represents the interaction between the two factors (so it should not be “+”). This style of table is the most commonly used means to report statistical outcomes for factorial design experiments so it should be easily interpretable by most readers. However, we did add some additional clarifying language in the table legend as suggested, which should help some in case there is confusion:

“Insecticide × NCF denotes statistical interaction between the two main effects.”

“Also, there are some significant results that have no colour shading.”

Response: There were a few main effects that accidentally had no shading; these were merely oversights that are now fixed. Thank you for catching these! There are also a few interactions that are significant (queen weight and worker larvae). We intentionally left these unshaded since it only works for main effects, i.e., there’s no clear increase/decrease that can describe an interaction effect. We now note this in the figure legend:

“Significant ($P < 0.05$) and marginally significant ($P < 0.07$) insecticide and non-crop forage effects are bolded for emphasis. Yellow and red shading denote increases and decreases, respectively, of the response variable to treatments that were significant for main effects (note a few interactions were also significant; left unshaded).”

“It would actually be really good to know the direction of all the results, irrespective of significance (particularly as some are close to significance) – would that be possible?”

Response: We did this for the 3 variables with p-values ranging from 0.0561 to 0.0628 since these are very close to significance. However, doing it for all results, even ones that are not close to the 0.05 threshold would be very unusual so we only added color for the noted marginally significant effects. The raw data (means) are shown in the supplementary table so if readers are curious about directionality, that information is easily accessible to them.

“Figures 4 & 5: It would be good show the significance differences (of lack of) between the NCF (-) and NCF (+) bars as well.”

Response: Unfortunately, we cannot add this information directly to the figures since it would be statistically incorrect. In factorial design analyses, you are only justified in doing pairwise comparisons for factor A within each level of factor B (in this case – independently assessing NCF separately for Insecticide +/-) when the interaction effect is statistically significant, which was not the case for either of these variables. Only the main effect is significant for insecticide. Instead, we added a sentence to the figure legend noting the lack of overall NCF significance, which should serve the same purpose:

“Insecticide use significantly reduced activity, whereas non-crop forage had no significant effect.”

“Insecticide use significantly reduced fruit set, whereas non-crop forage had no significant effect.”